

# P5CR1 protein expression and the effect of gene-silencing on lung adenocarcinoma

Yang She[1], Aiyou Mao[1], Feng Li[2] and Xiaobin Wei[1]

[1] Clinical Laboratory, Central South University Xiangya School of Medicine Affiliated Haikou Hospital, Haikou City, Hainan Province, China
[2] Clinical Laboratory, Hainan Provincial People's Hospital, Haikou City, Hainan Province, China

## ABSTRACT

The present study aimed to investigate the expression of pyrroline-5-carboxylate reductase 1 (P5CR1) protein in lung adenocarcinoma and paracancerous tissues and to explore the effect of silencing the encoding gene *PYCR1* on the proliferation, migration, invasion, and cisplatin sensitivity in lung adenocarcinoma cells, thereby providing a novel therapeutic target for the treatment of the disease. Immunohistochemistry staining was used to detect the P5CR1 protein expression in lung adenocarcinoma and paracancerous tissues, and statistical analysis evaluated the correlation between P5CR1 protein expression and gender, age, tissue part, or pathological grade. The CCK8 assay was performed to detect the proliferation and cisplatin sensitivity, while the effect of *PYCR1* on the migration and invasion of lung adenocarcinoma cells was detected by scratch test and transwell chamber assay. The findings demonstrated that the P5CR1 protein expression was significantly elevated in lung adenocarcinoma tissues and correlated with the pathological grade, whereas no significant correlation was established between the protein expression and gender, age, or tissue part. Furthermore, after *PYCR1* gene silencing, the proliferation and invasion were significantly suppressed, while the sensitivity to cisplatin was significantly enhanced. Therefore, it can be speculated that the *PYCR1* gene affects the biological behavior of lung adenocarcinoma and cisplatin resistance, serving as a potential therapeutic target for lung adenocarcinoma.

## INTRODUCTION

According to the Global cancer statistics 2012, lung cancer is the primary cause of cancer mortality, and non-small cell lung cancer (NSCLC) accounted for approximately 80–90% of lung cancer deaths worldwide (*Torre et al., 2015*). As an important pathological type of NSCLC, the incidence of lung adenocarcinoma is increasing year by year, and has even surpassed lung squamous cell carcinoma as the most important pathological type of NSCLC (*Lortet-Tieulent et al., 2014*). In China, approximately 0.3 million individuals are diagnosed with lung cancer, and of these about 0.25 million succumb to mortality annually (*Chen et al., 2016*). Lung cancer is associated with high incidence and mortality, causing heavy burden to the society.

Corresponding author
Xiaobin Wei,
westleywei1972@163.com

A majority of the lung cancer patients do not exhibit conspicuous symptoms or only have nonspecific symptoms, such as cough, which could be easily ignored. Therefore, most patients are diagnosed only during distant metastasis and are ineligible for surgery. The detection of early lung cancer is difficult. Currently, cisplatin-based combination chemotherapy is the primary therapeutic regimen for advanced NSCLC (*Willers et al., 2013*). However, cisplatin alone easily elicits drug resistance. According to the American Cancer Society, 90% of the patients succumb to cancer-related deaths due to drug resistance (*Lee & Dutta, 2007*). Several of studies (*Bersanelli et al., 2016*; *Chufan, Sim & Ambudkar, 2015*; *Pasquier, 2015*) have found that the mechanism underlying the tumor resistance to chemotherapeutic drugs is associated with complicated and varied cellular signaling pathways, in which, mutations in one or multiple molecules cause tumor cell invasion and drug resistance. Therefore, exploring the precise molecular mechanism underlying the physiological behavior and drug resistance of lung adenocarcinoma is imperative.

The P5CR1 protein, which is encoded by the housekeeping gene *PYCR1*, reduces the pyrroline-5-carboxylate (P5C) to proline in the presence of NADH and regulates the tricarboxylic acid cycle, the urea cycle, and the pentose phosphate metabolic pathway (*Hu, Phang & Valle, 2008*; *Phang, Liu & Hancock, 2013*). In addition, the P5CR1 also regulates certain biological behaviors of tumor cells. *PYCR1* is one of the most commonly over-expressed metabolic genes in 1981 tumor samples spanning 19 types of cancers. This suggested that the cancer cells may be addicted to high levels of *PYCR1* to sustain cell growth and provide oxidative stress resistance for cell survival (*Nilsson et al., 2014*). In recent years, in-depth analysis of malignant melanoma, prostate cancer, breast cancer, and other tumors further confirmed the importance of *PYCR1* in the occurrence and development of tumors (*Cai et al., 2018*; *Ding et al., 2017*; *Ye, Wu & Wang, 2018*; *Zeng et al., 2017*). *Cai et al. (2018)* analyzed the data regarding the expression of *PYCR1* in lung cancer in the Oncomine platform and found that the *PYCR1* mRNA expression was elevated in lung adenocarcinoma tissues. This study further unveiled that *PYCR1* could promote the proliferation of NSCLC cells and inhibit its apoptosis, which is subsequently related to the regulation of the expressions of cyclin D1, Bcl-2 and Bcl-xl in the regulation of cell cycle of *PYCR1* gene (*Cai et al., 2018*). While other studies showed that *PYCR1* is regulated by microRNA-488, which can activate the p38 MAPK pathway to promote the proliferation and apoptosis of NSCLC (*Wang et al., 2019*). Nonetheless, the role of *PYCR1* in invasion and resistance to cisplatin of lung adenocarcinoma migration, has not yet been reported.

The present study collected specimens of lung adenocarcinoma tissue and paracancerous tissue. Immunohistochemistry (IHC) staining detected the differential expression of P5CR1 protein between cancer and paracancerous tissues, and statistical analysis evaluated the correlation between P5CR1 protein levels and the relevant clinical variables. Moreover, we established *PYCR1*-silenced lung adenocarcinoma cells to evaluate the changes in cell proliferation, migration, invasion, cisplatin sensitivity. This study explored the role of *PYCR1* in the physiological behavior of lung adenocarcinoma and the potential mechanism underlying the tumor cell resistance to cisplatin, which provided novel insights into lung adenocarcinoma to formulate strategies for the treatment.

## MATERIALS AND METHODS

### IHC detection of P5CR1 protein

After the approval of the ethics committee of Haikou People's Hospital with approval number SC20180050 and the participant informed consent signed by the patient, we collected the lung adenocarcinoma tissues of the patients who underwent lobectomy or segmentectomy in our hospital from April 2018 to September 2018, and they had not received radiotherapy and chemotherapy before operation. The tissue sections were deparaffinized, heated in boiling 0.01 M sodium citrate buffer (pH 6.0) for 20 min in a microwave for antigen retrieval, and cooled to room temperature, followed by soaking in distilled water for 10 min before blocking with 10% serum in tris buffered saline (TBS) for 30 min. Subsequently, the sections were probed with the primary antibodies overnight, followed by the appropriate secondary antibodies for 60 min at room temperature. Next, Vulcan Fast Red Chromogen Kit2 was applied for 15 min before ceasing the reaction. Finally, diaminobenzidine (DAB) reagent was added and incubated until the appearance of a light yellow color, followed by addition of distilled water to stop the reaction and hematoxylin staining for 30. The stained tissue underwent dehydration by gradient concentrations of ethanol and was air dried before observation under a microscope for image analysis.

### Cell culture

Lung adenocarcinoma cells NCI-H1299 and A549 were purchased from the Shanghai Cell Institute of Chinese Academy of Sciences. The cells seeded in a six-cm tissue culture plate with three mL complete media and incubated at 37 °C and 5% $CO_2$. The cells in the exponential phase of proliferation were used for subsequent experiments.

### Construction of *PYCR1*-silencing recombinant lentivirus vector and virus infection

According to the *PYCR1* gene sequences in the NCBI database, the siRNA sequences were designed as follows: 5′-CCGGGAGGGTCTTCACCCACTCCTACTCGAGT AGGAGTGGGTGAAGACCCTCTTTTTG-3′ and 5′-GAATTCAAAAAGAGGGTC TTCACCCACTCCTATCTCGAGTAGGAGTGGGTGAAGACC-3′, independently. The lyophilized shRNA pair sequences were solubilized in the annealing buffer and heated for 15 min at 90 °C before naturally cooling to room temperature to form the double-stranded (ds) DNA. T4 DNA ligase was used to ligate the empty vector with annealed dsDNA, in between *EcoRI* and *AgeI*. Then, the recombinant vector construct was transfected into the competent cells. Isolated colonies were subjected to the identification by PCR. The identified positive colonies were individually cultured for 12–16 h in LB media with appropriate selection antibodies at 37 °C before sequencing. The sequencing reads were aligned against the target gene sequences, and the recombinant vector harboring the correct sequences were used for subsequent experiments. The plasmid was extracted using the EndoFree Midi Plasmid Kit from Tiangen. Then, the empty vector GV115 or recombinant vector coming was co-transfected into 293T cells with virus packaging help plasmids Helper 1.0 and Helper 2.0. In 48–72 h following transfection, the viruses were collected, concentrated, and purified, and the virus titter was measured

by the fluorescence method. Subsequently, NCI-H1299 and A549 cells were infected with one of the two virus vectors to establish the control vector-carrying or *PYCR1*-silencing lung adenocarcinoma cells, and the transfection efficiency of the vector control group (NC group) and *PYCR1*-silencing group (KD group), respectively, was measured by fluorescence microscopy.

## RT-PCR

NCI-H1299 and A549 cells in the KD/NC group and non-transfected parent cells (blank control group/CON group) were collected. The total RNA of individual groups was extracted using the TRIzol reagent, followed by the estimation of RNA concentration and quality with Nanodrop 2000/2000C Spectrophotometer. Subsequently, reverse transcription to synthesize the cDNA was performed with the Promega M-MLV Kit, followed by *RT-PCR*. *PYCR1* primers were: forward 5′-GGCTGCCCACAAG ATAATGGC-3′; reverse 5′-CAATGGAGCTGATGGTGACGC-3′. GAPDH primers were used to amplify the internal gene: forward 5′-TGACTTCAACAGCGACA CCCA-3′; reverse 5′-CACCCTGTTGCTGTAGCCAAA-3′. A two-step *RT-PCR* was performed to detect the gene expression levels in the *PYCR1*-silencing group in comparison with the vector control and blank control groups.

## Western blotting

The cells in the logarithmic growth phase were taken, washed twice with phosphate buffer saline (PBS), treated twice with lysis buffer, and then were pre-cooled for cleavage. Th cells were scraped off and transferred into an EP tube. After 10–15 mins of ice lysis, the cells were broken by ultrasound (200 W four times, 5 s each time, with 2 s interval). The protein concentration was determined by BCA after centrifugation for 15 mins. The protein concentration of each sample was then adjusted to two ug/mµ L. Gel electrophoresis of the samples was then performed at 120 mA for 1 h. After that, the protein was transferred onto the polyvinylidene fluoride (PVDF) membrane by using a transfer electrophoresis device at 4 °C and 300 mA for 150 min at constant current. The PVDF membranes were sealed at room temperature for 1 h using tris buffered saline tween (TBST) solution containing 5% skimmed milk. The first antibody was diluted by blocking solution, and then incubated with the blocked PVDF membrane at room temperature for 2 h. After that, the membrane was washed with TBST four times, each time for 8 mins. Next, the PVDF membrane was incubated at room temperature for 1.5 h and then washed with TBST for four times, each time for 8 mins. After that, X-ray development was done and then analysis.

## CCK8 assay to detect the proliferation

The proliferating cells were digested with trypsin, followed by suspension in complete media. The cells were seeded at a density of 2,000 cells/well in 100 µL media in the triplicate/group, in a total of five 96-well plates. Starting from the following day, 10 µL CCK-8 reagent was added into each well an incubated at two to four for measuring the OD value at 450 nm wavelength using a microplate reader.

## Scratch test to detect cell migration

$5 \times 10^4$ cells/well in 100 μL were seeded in the 96-well plate. Upon 90% confluency, the media were replaced with that containing low concentration serum. The scratch was generated by positioning a scratch tester on the lower center of each well and gently pushing forward. Then, the cells were rinsed two to three times with serum-free media before culturing in low concentration serum (0.5% fetal bovine serum (FBS)) and photographed. Subsequently, the cells were cultured in an incubator at 37 °C and 5% $CO_2$ and photographed at 8 and 24 h under a fluorescence microscope. The migration rate was calculated based on the post-scratch images.

## Transwell chamber system to detect the invasion

Cells suspensions were prepared in serum-free media at a density of $10^5$ cells/well. Then, the transwell chamber system was placed on a new 24-well plate, and the upper chamber was filled with 100 μL serum-free media before placing it in the 37 °C incubator for 1 h. Subsequently, 100 μL media in the upper chamber were replaced with 100 μL cell suspension in serum-free media, while the lower chamber was filled with 600 μL culture media containing 30% FBS, followed by incubation of the whole system at 37 °C for 24 h. Next, the culture media was removed from the upper chamber, and the non-migratory cells in the chamber were removed using the cotton swabs. Giemsa stain was applied on the opposite surface of the membrane to stain migrated cells for 3–5 min. Finally, the chamber was soaked, rinsed several times, air-dried, and photographed under the microscope.

## CCK-8 assay to detect cisplatin sensitivity

The cells were trypsinized, resuspended in complete culture media, and seeded in the 96-well plate at a density of 4,000 cells/well in 100 μL. Then different concentrations of the drug were added. Based on the study design, the drug intervention lasted for 48 h. After that, 10 μL CCK-8 reagent was added into each well for 2–4 h without refreshing the media before measuring the OD value at 450 nm using a microplate reader.

## Statistical analysis

SPSS 21.0, Photoshop, and GraphPad Prism 5 were used for the statistical analysis of the experimental data. Quantitative data are expressed as mean ± SD. The independent sample $t$-test was used to compare the two groups, and the analysis of variance was used to compare the variances between the groups. $P < 0.05$ indicated the statistically significant difference.

# RESULTS

## P5CR1 protein expression in lung adenocarcinoma vs. paracancerous tissue and its correlation with clinical variables

Immunohistochemistry detection demonstrated a significant increase in the expression of P5CR1 protein in 28 cases of lung adenocarcinoma tissues than that in 27 adjacent tissues ($P < 0.001$; Figs. 1A and 1B). The correlation of P5CR1 protein expression with gender, age or pathological grade, respectively, was analyzed by independent samples

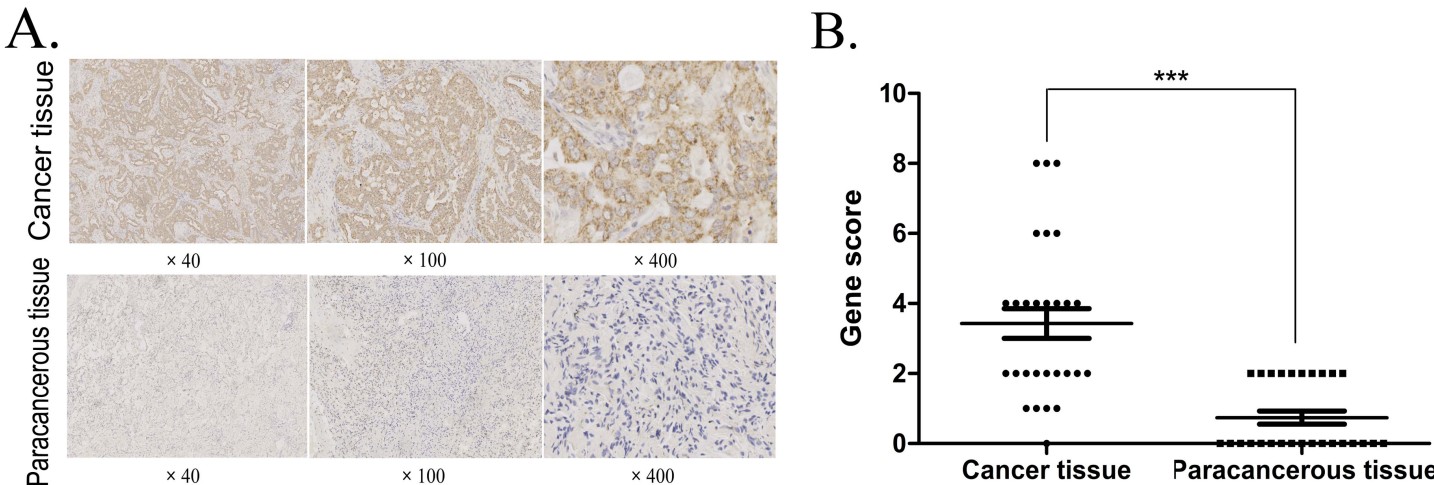

**Figure 1  IHC detection of P5CR1 protein expression.** (A) The P5CR1 antibody binds to the P5CR1 protein in the cell pellet and is stained brown. Scoring system for immunohistochemical staining: The percentage of positive cells was determined on a scale of 0–4 points, where 0 points, no positive cells; 1 point, the percentage of positive cells was less than 10%; 2 points, the percentage of positive cells was greater than 10% but less than 50%; 3 points, the percentage of positive cells was greater than 50% but less than 80%; and 4 points, the percentage of positive cells is greater than 80%. The staining intensity was rated on a scale of 0–3 points by negative, weak, medium, and strong staining, respectively. The final score is multiplied by the two metrics; (B) The data are presented as mean and SD of P5CR1 protein in lung adenocarcinoma ($n = 28$) and adjacent tissues ($n = 27$); (***$p < 0.001$).

$t$-test; The expression level of P5CR1 protein in different lesions was analyzed by variance analysis. The results revealed that the expression of P5CR1 in lung adenocarcinoma was significantly higher in grade IIB-IIIB group than in grade IA-IIA group ($P < 0.001$), but no significant correlation was detected with gender, age, or tissue part ($P > 0.05$; Table 1).

### Construction of *PYCR1*-silencing of lung adenocarcinoma cells

Based on the *PYCR1* gene sequence in the NCBI database, the siRNA sequence GGGTCTTCACCCACTCCTA was designed as shRNA and integrated into the vector. The subsequent sequencing confirmed that the recombinant virus vector carried the correct sequences (Fig. 2A). The virus vector was then infected into lung adenocarcinoma NCI-H1299 and A549 cells. The transfection efficiency was detected to be >80% using fluorescence microscopy (Figs. 2B and 2C). Next, the gene silencing efficiency and protein expression of NCI-H1299 and A549 cells were verified by RT-PCR and WB, respectively. Using independent sample $t$-test analysis, the results showed that the *PYCR1* gene silencing efficiency of NCI-H1299 and A549 cells was 92.1% and 82.4%, respectively (both $P < 0.01$; Fig. 2D), and the expression of P5CR1 protein was significantly down-regulated when compared with the empty vector group (both $P < 0.01$; Figs. 2E and 2F). Hitherto, *PYCR1*-silencing lung adenocarcinoma cells were established successfully.

### *PYCR1*-silencing inhibited the cell proliferation of lung adenocarcinoma cells

Lung adenocarcinoma NCI-H1299 and A549 cells in *PYCR1*-silencing/vector control/blank control groups were cultured for 24 h, and then cell proliferation in each group was

**Table 1 Correlation of P5CR1 protein level with clinical variables.**

| Parameters | n | P5CR1 protein | |
|---|---|---|---|
| | | Expression | P |
| Gender | | | |
| Male | 16 | 3.002 ± 2.296 | 0.397 |
| Female | 12 | 3.753 ± 2.266 | |
| Age | | | |
| ≤60 | 14 | 3.361 ± 2.341 | 0.873 |
| >60 | 14 | 3.504 ± 2.279 | |
| Pathological grading | | | |
| IA-IIA | 14 | 1.933 ± 1.072 | 0.000 |
| IIB-IIIB | 14 | 4.937 ± 2.165 | |
| Tissue part | | | |
| R+U | 8 | 4.125 ± 2.416 | 0.798 |
| R+L | 8 | 3.125 ± 3.091 | |
| L+U | 6 | 3.000 ± 1.549 | |
| L+L | 6 | 3.333 ± 1.633 | |

Note:
n, number of patients; P, significant level; R+U, right upper lobe; R+L, right lower lobe; L+U, left upper lobe; L+L, left lower lobe.

measured using CCK8 reagent for five consecutive days. Independent sample $t$-test was performed to analyze the experimental data. The results showed that the OD450 of *PYCR1*-silenced group was lower than that of vector control group on days 2–5 in A549 and NCI-H1299 cells (both $P < 0.01$; Figs. 3A and 3B). This indicated that *PYCR1*-silencing inhibited the proliferation of lung adenocarcinoma cells.

## *PYCR1*-silencing has no effect on the migration ability of lung adenocarcinoma cells

Lung adenocarcinoma NCI-H1299 and A549 cells in *PYCR1*-silencing/vector control/blank control groups were inoculated on 96-well plate, respectively. Scratch size was measured under inverted fluorescence microscope at 0, 8, and 24 h after the scratch. The width of the scratch area was determined by the W value derived from the Photoshop Software, which was further analyzed by independent sample $t$-test. The results demonstrated significant differences between the vector control group and the *PYCR1*-silenced lung adenocarcinoma NCI-H1299 cells after 24 h post-scratch ($P < 0.05$; Figs. 3C and 3D), whereas the variation in the migration rate was <20%. This further indicated the lack of positive results. Compared to the vector controls, *PYCR1*-silencing lung adenocarcinoma A549 cells showed no significant difference in the migration ability at 24-h post-scratch ($P > 0.05$; Figs. 3C and 3D).

## *PYCR1*-silencing suppresses the cell invasion ability of lung adenocarcinoma cells

Lung adenocarcinoma NCI-H1299 and A549 cells in *PYCR1*-silencing/vector control/blank control groups were inoculated into the upper chamber of Transwell

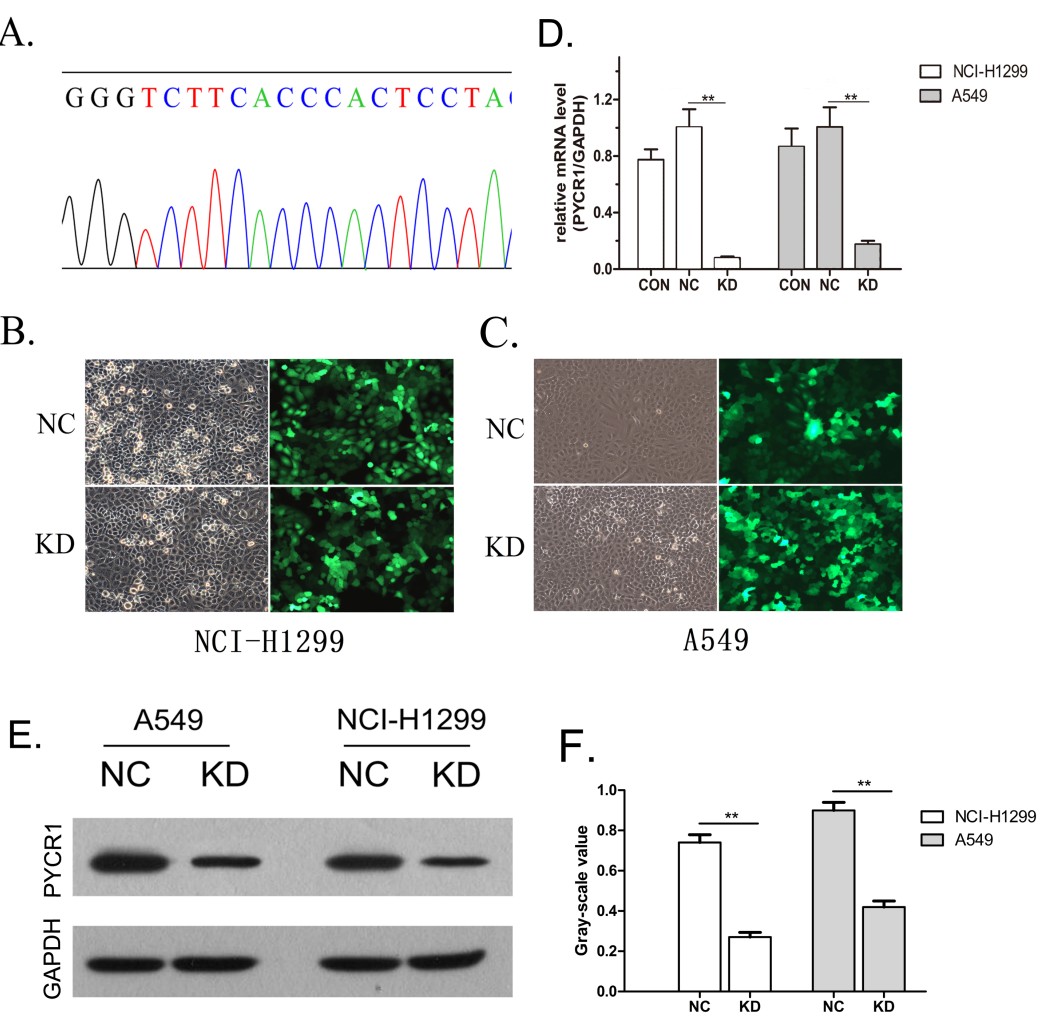

**Figure 2 Construction of *PYCR1*-silenced lung adenocarcinoma cells.** (A) Sequencing of siRNA to confirm the sequence. (B and C) Fluorescence microscopy for the detection of transfection efficiency. Positive cells presented green fluorescence at 100× magnification. (D) RT-PCR for the detection of *PYCR1* expression. mRNA expression levels in KD group were significantly decreased compared with NC groups. Data are presented as mean ($n = 3$) and SD. (E) Western blotting was used to detect the expression of P5CR1 protein. (F) Histogram showed the expression of P5CR1 protein. The expression level of P5CR1 protein in KD group was significantly decreased compared with NC groups. Data are presented as mean ($n = 3$) and SD. (**$P < 0.01$).

chamber. After 24 h of incubation at 37 °C, Giemsa staining was performed to detect the number of cells migrating to the lower chamber and adhering to the bottom of the polycarbonate membrane. Independent sample *t*-test was used to analyze the experimental results. Transwell assay demonstrated that the number of lung adenocarcinoma NCI-H1299 and A549 cells migrating to the inferior cavity in the *PYCR1* silencing group was significantly reduced when compared with the vector control group (both $P < 0.01$; Figs. 4A–4C), indicating that *PYCR1*-silencing led to a significant reduction in the invasiveness of lung adenocarcinoma NCI-H1299 and A549 cells.

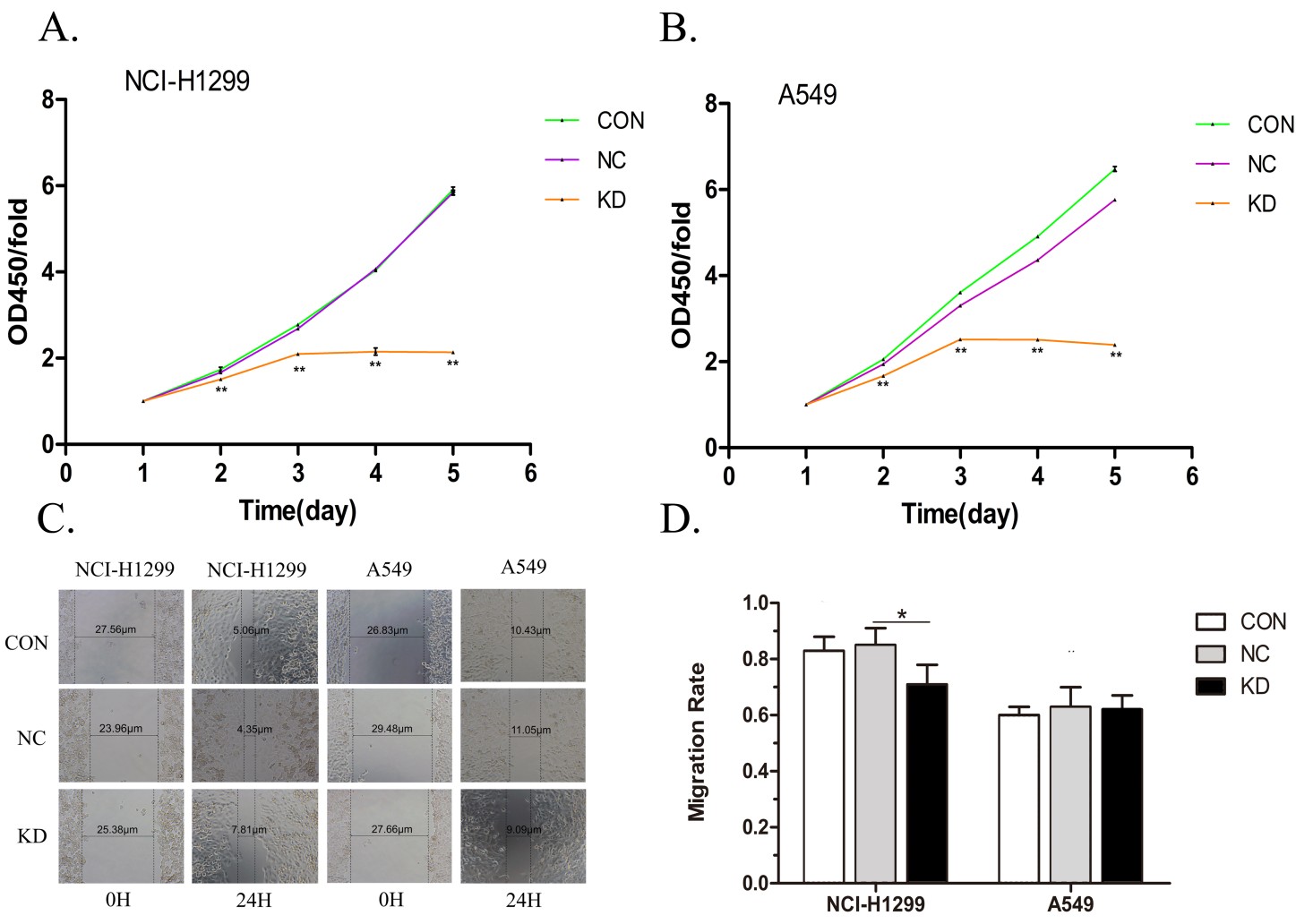

**Figure 3 Detection of proliferation and migration ability of lung adenocarcinoma cells.** (A and B) Three differently colored lines were used to simulate changes in the cell viability over a period in different groups of lung adenocarcinoma cells. The proliferation of the KD group was significantly inhibited compared with the NC groups. Data are represented as mean ($n = 5$) and SD. (C) The distances marked in the figure represent the size of the scratches measured by inverted fluorescence microscopy at 0 and 24 h in the three different groups of NCI-H1299 and A549 cells. The images were captured at 40×. (D) This histogram represents the cell migration ability. *PYCR1*-Silencing had no effect on migration of NCI-H1299 and A549 cells in lung adenocarcinoma. Data are represented as mean ($n = 3$) and SD. (*$P < 0.05$; **$P < 0.01$).

## *PYCR1*-silencing enhances the sensitivity of lung adenocarcinoma cells to cisplatin

Lung adenocarcinoma NCI-H1299 and A549 cells in *PYCR1*-silenced/vector control/blank control groups were treated with different concentrations of cisplatin, and the rate of inhibition on the proliferation was detected using the CCK8 assay. Independent sample *t*-test was used to analyze the experimental results. The cisplatin stimulation with 1, 2, 4, 6, 8, 10, 16 mg/L significantly inhibited the NCI-H1299 and A549 cell proliferation in the *PYCR1*-silenced group as compared to the vector control group (both $P < 0.01$; Figs. 4D and 4E). These results indicated that the *PYCR1*-silencing elevated cisplatin sensitivity in lung adenocarcinoma cells.

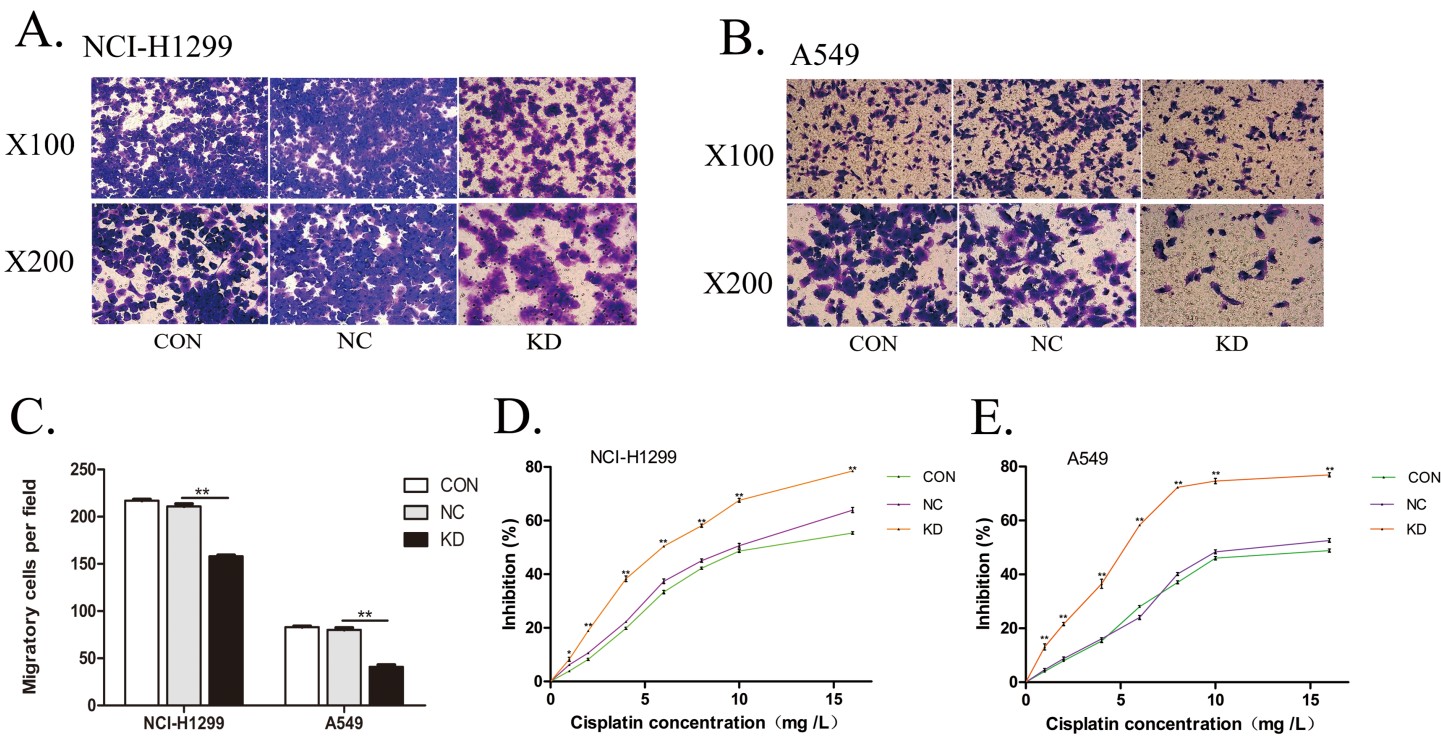

**Figure 4 Detection of invasion and cisplatin resistance in lung adenocarcinoma cells.** (A and B) Lung adenocarcinoma NCI-H1299 (A) and A549 (B) cells that penetrate the polycarbonate membrane in the lower chamber are stained purple. (C) This histogram represents the cell invasion ability of lung adenocarcinoma. Compared with NC groups, the number of migration to the lower chamber in KD group was significantly reduced. Data are represented as mean ($n = 3$) and SD. (D and E) Three different color lines were used to simulate the effects of different concentrations of cisplatin on cell proliferation in each group. The proliferation of the KD group was significantly inhibited compared with the NC groups. Data are represented as mean ($n = 3$) and SD. (*$P < 0.05$; **$P < 0.01$).

## DISCUSSION

5-pyrrolinecarboxylic acid reductases (P5CRs) are housekeeping proteins that are widely present in the prokaryotic cells as well as eukaryotic cells and contain three isoenzymes: P5CR1, P5CR2, and P5CR3. The P5CRs-proline metabolic cycle reaction is not only closely associated with the body's tricarboxylic acid cycle, urea cycle, and pentose phosphate pathway, but it also plays a regulatory role in the proliferation and apoptosis of cells, as well as, skin, bone, brain, and other tissues' development and in the process of occurrence and development of oral, head and neck, lymphoma, and other tumors (*Liu et al., 2005*; *Phang et al., 2012*; *Reversade et al., 2009*). The human P5CR1 gene, *PYCR1* is located on chromosome 17q25.3 and is five kb in length including seven exons and eight introns (*De Ingeniis et al., 2012*). P5CR1 is involved in amino acid synthesis and metabolism, and promoted the occurrence and development of malignant melanoma, prostate cancer, lung cancer, breast cancer, and other tumors (*Cai et al., 2018*; *Ding et al., 2017*; *Ye, Wu & Wang, 2018*; *Zeng et al., 2017*). Currently, some studies suggested that *PYCR1* may regulate the biological behavior of tumors by regulating the transport of NADH reduction equivalent from cytoplasm to mitochondria, further affecting the level of intracellular reductant synthesis and the sensitivity to oxidative damage (*Gao et al., 2009*).

Also, P5CRs may reduce the production of reactive oxygen species in the mitochondria through oxidative respiratory chain process involved in NADP+ to increase the survival rate of tumor cells (*Phang et al., 2012*).

The present study investigated the expression of P5CR1 protein in lung adenocarcinoma and the effect of encoding gene-silencing on the physiology of the cells. P5CR1 protein was highly expressed in lung adenocarcinoma tissues, and its level was found that to be positively correlated with the pathological grade. Thus, we speculated that the *PYCR1* gene might be related to the specific physiological characteristics of lung adenocarcinoma. Moreover, to probe the effect of differential expression of the gene on the proliferation, migration, invasion, and drug sensitivity of lung adenocarcinoma, we successfully established the *PYCR1*-silenced lung adenocarcinoma cells. Furthermore, CCK8 assay demonstrated that *PYCR1*-silencing significantly inhibited the cell proliferation in lung adenocarcinoma. The cell migration was detected by the scratch test, and the silencing of the *PYCR1* gene did not affect the migration of lung adenocarcinoma cells significantly. Next, the lung adenocarcinoma cell invasion analysis by Transwell chamber system demonstrated that *PYCR1*-silenced lung adenocarcinoma cells showed a significant reduction in the invasion ability. In order to explore the role of *PYCR1* in tumor cell resistance to cisplatin, lung adenocarcinoma cells in *PYCR1*-silenced/vector control/blank control groups were treated with cisplatin at different concentrations, followed by detection of inhibition of the proliferation using the CCK8 assay. The results demonstrated the *PYCR1*-silencing led to a rise in cisplatin sensitivity in lung adenocarcinoma cells.

Although some studies explored the role of *PYCR1* in the proliferation of human lung squamous cell carcinoma H1703 cells and human lung adenocarcinoma SPC-A1 cells (*Cai et al., 2018*), the correlation of *PYCR1* gene with cell migration, invasion, and cisplatin sensitivity in lung adenocarcinoma was not examined. The current study selected the lung adenocarcinoma cell lines HCI-H1299 and A549. The results demonstrated that *PYCR1* promoted lung adenocarcinoma cell proliferation, which was in agreement with the above study on human lung adenocarcinoma SPC-A1 cells. This study is the first to explore the effect of *PYCR1* gene on migration, invasion, and cisplatin sensitivity of lung adenocarcinoma, and confirmed that *PYCR1* gene promoted the invasion and caused cisplatin resistance. Although the results of the study indicated that silencing of *PYCR1* gene showed no significant effect on the migration of lung adenocarcinoma cells, while significantly promoted the invasiveness. This might be due to the fact that the *PYCR1* has little effect on simple horizontal movement of lung adenocarcinoma cells, but can regulate some substances, which led to the remodeling and degradation of extracellular matrix, thereby enhancing the invasiveness of cells. *PYCR1* converts the P5C into proline and increases NAD+ production. Studies have shown that elevated levels of NAD+ can promote glycolysis of lactic acid through glyceraldehyde 3-phosphate dehydrogenase and lactate dehydrogenase, resulting in increased lactic acid content (*Tan et al., 2013*; *Yamamoto, Inohara & Nakagawa, 2017*). Excessive lactic acid activates monocarboxylic acid transporter 1, causing intracellular lactic acid efflux. Extracellular low pH value provides a good microenvironment for the release and activation of matrix

metalloproteinases, tumor cathepsin B and so on. Activation of these enzymes can promote extracellular matrix degradation, and remodeling, leading to tumor metastasis (*Dhup et al., 2012*). Therefore, we speculated that *PYCR1* enhanced the invasion of lung adenocarcinoma cells and this might be related to its enhancement of glycolysis and lactic acid production in tumor cells. This study demonstrated that silencing the *PYCR1* gene can enhance the sensitivity to cisplatin that rendered *PYCR1* as a promising candidate target for the treatment of lung adenocarcinoma. However, the *PYCR1*-overexpression study has not been conducted and remain only at the cellular level. Thus, the future experiments will study the effect of *PYCR1*-overexpression on the physiological behavior of lung adenocarcinoma in the corresponding cellular and animal experiments.

P5CR1 is an internal protein close to the mitochondrial inner membrane and plays a key role in the regulation of substance/energy metabolism. Therefore, we consider that the *PYCR1* gene may influence the biological behavior of lung adenocarcinoma cells by regulating cell energy/material metabolism, and this is also our next research direction.

## CONCLUSIONS

This study found that the *PYCR1* gene is highly expressed in lung adenocarcinoma cells, and is positively correlated with the pathological grade of lung adenocarcinoma. The *PYCR1* gene promotes the physiological behaviors such as proliferation and invasion of lung adenocarcinoma cells. Furthermore, *PYCR1* plays a critical role in resistance of lung adenocarcinoma to cisplatin, which might be a potential therapeutic target for lung adenocarcinoma, thereby providing theoretical data for the chemotherapy of the disease.

## ACKNOWLEDGEMENTS

We greatly appreciate Denggao Hang  and Xin Gao for experimental image processing and the Central Laboratory of Haikou People's Hospital for providing the experimental platform.

### Funding

This work was supported by the Finance Science and Technology Project of Hainan Province (No. ZDYF2018132). The funders had no role in study design, data collection and analysis, decision to publish, or preparation of the manuscript.

### Grant Disclosures

The following grant information was disclosed by the authors:
The Finance Science and Technology Project of Hainan Province: ZDYF2018132.

### Competing Interests

The authors declare that they have no competing interests.

## Author Contributions

- Yang She conceived and designed the experiments, performed the experiments, analyzed the data, contributed reagents/materials/analysis tools, prepared figures and/or tables, authored or reviewed drafts of the paper.
- Aiyou Mao performed the experiments.
- Feng Li performed the experiments.
- Xiaobin Wei conceived and designed the experiments, authored or reviewed drafts of the paper, approved the final draft.

## Human Ethics

The following information was supplied relating to ethical approvals (i.e., approving body and any reference numbers):

The Haikou People's Hospital granted Ethical approval to carry out the study within its facilities (Ethical Application Ref: SC20180050).

## Data Availability

The raw measurements are available in Files S1–S6.

## Supplemental Information

Supplemental information for this article can be found online at http://dx.doi.org/10.7717/peerj.6934#supplemental-information.

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
