# Peer review of "P5CR1 protein expression and the effect of gene-silencing on lung adenocarcinoma"

_PeerJ, doi:10.7717/peerj.6934_

## Round 0.1 · original submission · Major Revisions

Your article has been reviewed by experts in the field and we request that you make major revisions before it is processed further. (1) Please provide explanation of knockdown of PYCR1 did not affect the migration of lung adenocarcinoma cells.(2) To provide the reproducibility of the data and their statistical significance.

Please also address all the other comments of the reviewers.

Reviewer 1 ·

Basic reporting

In the work of Cai et al., they have shown that “Overexpression of PYCR1 in lung cancers was further supported by data analysis from the Oncomine platform, which revealed higher PYCR1 mRNA expression levels in lung adenocarcinoma tissues.” It would be good to include this statement in the introduction and provide more rationale of studying PYCR1

Experimental design

*It would be better to mention a little more about the rationale to investigate the role of P5CR1 protein in lung adenocarcinoma.

*It didn't look like there was a significant difference in the results of wound healing experiment. (by independent sample t-test? it would be inappropriate if there are more than two groups) What post-hoc test was used? Please do not show any symbol if the p value is not < 0.05. Also, is there any difference in Fig. 3A? Which statistical methods was used?

*Please state how the authors conducted the statistical analysis of the difference for the CCK-8 assay.

Validity of the findings

*It is suggested to relate the results with other previous results, including different types of cancers so the readers could gain a general idea of the role of PYCR1.

*Please explain that why silencing of the PYCR1 gene did not affect the migration of lung adenocarcinoma cells.

Reviewer 2 ·

Basic reporting

In this manuscript Yang She and collaborators present data to demonstrate that PYCR1 gene affects the biological behavior of lung adenocarcinoma suggesting a potential therapeutic target for lung adenocarcinoma with this approach. Overall, this is an interesting study but as indicated below, several results need to be backed up.
1. The major concern is the need to share the information regarding to if PYCR1 protein levels are affected by PYCR1-silencing recombinant lentivirus in lung adenocarcinoma cells. In addition, shRNAs can have off-target effects.In general, multiple shRNA should be tested for consistent and reproducible evidence.
2. Figure 1A, 1B, and Table1, there is no indication the scoring system for immunohistochemical staining
3. In introduction provide a rationale between NSCLC and lung adenocarcinoma.
4. Please check the histology of NCI-H1299 cells (adenocarcinoma or large cell carcinoma).
5. To point out the reproducibility of the data and their statistical significance, this information should be integrated with the number of experiments with similar or overlapping results. Otherwise the means (+/-SD) of all the experiments performed should be reported along with the appropriate statistical analysis.
6. The paper requires careful editing by a native English speaking scientist.

Experimental design

no comment

Validity of the findings

no comment

---

## Round 0.2 · accepted · Accept

The reviewers have satisfied the critiques of the reviewers.

# Reviewer 1 ·

Basic reporting

no comment

Experimental design

no comment

Validity of the findings

no comment

Reviewer 2 ·

Basic reporting

No comment.

Experimental design

No comment.

Validity of the findings

No comment.

Additional comments

No comment.